# Characterization of Spinal Sensorimotor Network Using Transcutaneous Spinal Stimulation during Voluntary Movement Preparation and Performance

**DOI:** 10.3390/jcm10245958

**Published:** 2021-12-18

**Authors:** Alexander G. Steele, Darryn A. Atkinson, Blesson Varghese, Jeonghoon Oh, Rachel L. Markley, Dimitry G. Sayenko

**Affiliations:** 1Center for Neuroregeneration, Department of Neurosurgery, Houston Methodist Research Institute, 6550 Fannin Street, Houston, TX 77030, USA; agsteele@houstonmethodist.org (A.G.S.); datkinson@usa.edu (D.A.A.); bvarghese6@houstonmethodist.org (B.V.); joh@houstonmethodist.org (J.O.); rmarkley@houstonmethodist.org (R.L.M.); 2Department of Electrical and Computer Engineering, University of Houston, E413 Engineering Bldg 2, 4726 Calhoun Road, Houston, TX 77204, USA; 3College of Rehabilitative Sciences, University of St. Augustine for Health Sciences, 5401 La Crosse Avenue, Austin, TX 78739, USA

**Keywords:** spinal cord, spinal stimulation, corticospinal tract, functional connectivity, movement, sensorimotor networks, task dependence

## Abstract

Transcutaneous electrical spinal stimulation (TSS) can be used to selectively activate motor pools based on their anatomical arrangements in the lumbosacral enlargement. These spatial patterns of spinal motor activation may have important clinical implications, especially when there is a need to target specific muscle groups. However, our understanding of the net effects and interplay between the motor pools projecting to agonist and antagonist muscles during the preparation and performance of voluntary movements is still limited. The present study was designed to systematically investigate and differentiate the multi-segmental convergence of supraspinal inputs on the lumbosacral neural network before and during the execution of voluntary leg movements in neurologically intact participants. During the experiments, participants (N = 13) performed isometric (1) knee flexion and (2) extension, as well as (3) plantarflexion and (4) dorsiflexion. TSS consisting of a pair pulse with 50 ms interstimulus interval was delivered over the T12-L1 vertebrae during the muscle contractions, as well as within 50 to 250 ms following the auditory or tactile stimuli, to characterize the temporal profiles of net spinal motor output during movement preparation. Facilitation of evoked motor potentials in the ipsilateral agonists and contralateral antagonists emerged as early as 50 ms following the cue and increased prior to movement onset. These results suggest that the descending drive modulates the activity of the inter-neuronal circuitry within spinal sensorimotor networks in specific, functionally relevant spatiotemporal patterns, which has a direct implication for the characterization of the state of those networks in individuals with neurological conditions.

## 1. Introduction

In individuals with a spinal cord injury (SCI), many sub-functional neural connections between the brain and spinal cord can remain intact across the injury, despite a clinical diagnosis of “complete” loss of sensorimotor function [1,2,3,4]. These connections are not robust enough to drive clinically detectable function; however, they are capable of influencing the excitability of spinal sensorimotor networks below the lesion [5,6,7,8]. These spinal networks can be activated by electrical spinal neuromodulation [9,10,11,12]. Notably, when combined with rehabilitation, epidural (ESS) or transcutaneous (TSS) spinal stimulation can promote functional recovery and increase overall well-being [13,14,15,16,17]. Computational [18,19] and electrophysiological [20,21,22] studies have demonstrated that the stimulus pulses recruit spinal circuits [21,23] that are involved in the regulation of rhythm- and pattern-generating neural networks [24,25,26]. It has been proposed that spinal stimulation modulates the physiological states of the spinal cord below the lesion, enabling descending and sensory information processing to generate coordinated and robust motor outputs, even after chronic “complete” (most likely “incomplete”) paralysis [14,27,28,29]. The complex interactions between stimulus pulses, descending commands originating above the SCI (and passing through the lesion site), and ongoing afferent inputs, which produce functional movement, have yet to be characterized. Furthermore, our understanding of physiological states of the intact spinal cord in humans is still limited; quantitative characterization of the net effects of descending commands and interplay between the agonist and antagonist motor pools within the lumbosacral spinal segments during voluntary movements is needed. In patients, electrophysiological evaluation of supraspinal-spinal and spinal inter-neuronal networks may be useful in characterizing the extent of residual function, as well as in identifying the neuroplastic changes associated with improved motor performance after neurological injuries and disorders [30,31].

Electrically induced soleus or quadricep H-reflexes have been used in earlier studies to document changes in spinal motoneuronal excitability associated with processes in the spinal sensorimotor network during voluntary movement preparation and execution in healthy [32,33,34,35,36] and spastic [34] individuals. The H-reflex was shown to be facilitated about 80 to 400 ms before the onset of a movement in the contracting muscles, due to a decrease in presynaptic inhibition to Ia fibers terminating onto discharging motoneurons [32,33,34]. Interestingly, the H-reflex magnitude in a given muscle can also be modulated by activity in other muscles [34]. For instance, the soleus H-reflex increased prior to and during ipsilateral plantarflexion, and it decreased prior to and during ipsilateral dorsiflexion. Similar, although less pronounced, changes were observed prior to and during contractions of the contralateral muscles. In later experiments, Hultborn et al. [35] proposed that the presynaptic inhibition of Ia afferent terminals on motoneurons of contracting muscles was decreased, permitting Ia activity to contribute to the excitation of voluntarily activated motoneurons; whereas, the presynaptic inhibition of motoneurons of muscles not involved in the contraction was increased. It was concluded that the control of presynaptic inhibition of Ia fibers prior to the movements is supraspinal in origin and is organized to aid in achieving selectivity of muscle activation. At the same time, Hultborn et al. [35] pointed out that during movement, the Ia-lb discharge occurring from the contracting muscle(s) also activates presynaptic interneurons, and it therefore provides a peripheral source of modulation of presynaptic inhibition of the motoneurons of the muscles not involved in the contraction.

The complex interaction of spinal inter-neuronal circuitry with descending drive in the preparatory stage, and during the execution of voluntary movements (or movement attempts), is particularly intriguing in the context of neuromuscular rehabilitation after paralysis. As the first step, we sought to investigate the multi-segmental convergence of the descending drive on the lumbosacral neural networks in neurologically intact participants using double-pulse TSS. Spinally evoked motor potentials are reminiscent of the monosynaptic H-reflex [37,38], but owing to the convergence of the sensory fibers at the lumbosacral enlargement, stimulation delivered over the posterior roots entering the spinal cord can generate evoked potentials in multiple proximal and distal leg muscles, bilaterally and simultaneously. We hypothesized that descending input prior to and during voluntary movements would modulate activity of the inter-neuronal circuitry within spinal sensorimotor networks in specific, functionally relevant spatiotemporal patterns. To assess this hypothesis, we examined the effects of knee flexion and extension, as well as plantarflexion and dorsiflexion, on the amplitude of spinally evoked motor potentials in multiple leg muscles.

## 2. Methods

### 2.1. Participants

Thirteen neurologically intact adults (6 females, 7 males; height: 170.5 ± 9.6 cm, weight: 69.0 ± 8.7 kg, age: 26.5 ± 5.4 years old) were recruited to participate in this study. Specifically, the healthy volunteers between 21 and 70 years old, who declared to have no medical history or current diagnostic of, or therapy for, neurological or orthopedic disorders were invited to participate in the study. The information about the study was posted on the laboratory website and on the Texas Medical Center campus. Written informed consent to the experimental procedures, which were approved by the Houston Methodist Research Institute institutional review board (Study ID: Pro00019704), was obtained from each participant.

### 2.2. Experimental Setup

TSS was delivered to the skin over the lumbosacral spinal enlargement using a constant current stimulator DS8R (Digitimer Ltd., Hertfordshire, UK). Stimulation was administered using conductive self-adhesive electrodes (PALS, Axelgaard Manufacturing Co. Ltd., Fallbrook, CA, USA). The cathode (diameter: 5 cm) was placed at midline between the T12 and L1 spinous processes, and two oval anodes (size: 7.5 cm × 13 cm) were placed on the abdomen, symmetrical to the sagittal plane.

To confirm the sensory route (i.e., via the posterior roots) of the delivered spinal stimulation and the transsynaptic transmission of the stimuli on the motor pools of interest, we used double-pulse TSS [39,40,41]. Double-pulse TSS was delivered using two 1 ms biphasic square wave pulses with an inter-stimulus interval of 50 ms. Stimulation began at 30 mA and increased in increments of 5 mA until motor responses were observed in the lower limb muscles. The location of the electrode was adjusted in a rostro-caudal direction as needed to obtain responses in proximal and distal limb muscles with the minimum difference in motor threshold stimulation intensity.

The auditory conditioning cue was delivered using a square 250 ms tone burst at 3000 Hz with an intensity of 70 dB generated via the PowerLab data acquisition system (ADInstruments, New South Wales, Australia). For the tactile condition cue, electrical stimulation was delivered to the skin over the metatarsal region on the plantar surface of the left foot using a constant current stimulator DS8R (Digitimer Ltd., UK). Stimulation was administered using conductive self-adhesive electrodes (PALS, Axelgaard Manufacturing Co. Ltd., USA). Two electrodes (diameter: 3.2 cm) were placed over the fibular sesamoid (cathode) and the center of the medial longitudinal arch (anode). Stimulation was delivered using a single monophasic 500 µs square wave pulse. Tactile stimulation began at 2 mA and was increased in increments of 1 mA until the participant reported a distinct sensation when stimulation was applied. Stimulation did not produce visible contractions of plantar muscles.

Trigno Avanti wireless surface electromyography (EMG) electrodes (Delsys Inc., Natick, MA, USA; common-mode rejection ratio < 80 dB; size: 27 mm × 37 mm × 13 mm; input impedance: >1015 Ω/0.2 pF) were placed at eight different sites: longitudinally over the left and right vastus lateralis (VL), medial hamstrings (MH), tibialis anterior (TA), and the lateral portion of the soleus (SOL) muscles, as seen in Figure 1A. EMG data were amplified using a Trigno Avanti amplifier (Delsys Inc, Natick, MA.; gain: 909; bandwidth: 20 to 450 Hz) and recorded at a sampling frequency of 2000 Hz using a PowerLab data acquisition system (ADInstruments, New South Wales, Australia).

### 2.3. Experimental Procedure

The tests were performed in the supine position. The participants’ legs were placed in the Exolab apparatus (Antex Lab LLC, Moscow, Russia) supporting the hip, knee, and ankle joints at 155, 90, and 90 degrees, respectively. Force output during isometric left knee extension, left knee flexion, left dorsiflexion, and left plantarflexion was measured using calibrated, two-sided precision load cells with a capacity of 50 kg (Futek, Irvine, CA, USA). Measurements were recorded at a sampling frequency of 2000 Hz using a PowerLab data acquisition system (ADInstruments, New South Wales, Australia). Initially, double-pulse TSS were applied at a subthreshold intensity, then increased in 5 mA steps with three pulses delivered at each intensity and continued until the magnitude of spinally evoked motor potentials in all muscles were observed to plateau or be at maximum tolerated intensity. This allowed for the generation of recruitment curves for each muscle, which were used to identify stimulation intensities and supra-threshold values of responses for conditioning experiments [30,31].

Figure 1B outlines the following experimental conditions. First, ten repetitions of fast isometric contractions on the auditory cue “as quickly as possible” were performed to calculate the baseline reaction time. Participants were asked to perform a fast isometric contraction of the muscle being tested and produce the 5 to 10% maximum voluntary contraction force. Specifically, the participants were instructed to perform isolated contractions of the agonist muscles, while trying not to engage synergists and antagonists. For instance, during knee extension, the participants were asked to contract only the quadriceps muscles while maintaining the knee flexors and calf muscles relaxed. Next, double-pulse TSS were applied ten times at 5 s intervals, in a relaxed condition (baseline control). After that, auditory cues were given every 5 to 7 s with pseudorandom spacing between them to reduce the possibility of anticipation of the following cue. The participants were asked to perform the predetermined movement on the cue as quickly as possible, then return to rest. TSS was applied at a condition-test interval (CTI) of 50, 100, 150, 200, and 250 ms after the auditory cue was given. CTIs were randomized to reduce carry-over effects. Each movement task was repeated for a total of ten times per CTI. A final ten double-pulse TSS were done at 5 s intervals with no movement being performed (post-conditioned control). Finally, the participants were instructed to maintain the isometric contraction of the muscle being tested, producing the 5 to 10% maximum voluntary contraction force, while double-pulse TSS was applied 2 s after the start of isometric contraction. After 2 s of the isometric contraction, a cue was given for the participant to relax the muscles. The isometric contraction test was performed ten times per movement condition with a minimum of 5 s of rest between repetitions. Example muscle responses from the left SOL obtained at different CTIs are shown in Figure 1C.

The experiment was then repeated using the tactile cue, except for the isometric contraction portion of the protocol.

### 2.4. Statistical Analysis

Statistical analysis was performed using the open-source software RStudio version 1.4.1717 (RStudio PBC, Boston, MA, USA). The minimum sample size of the study was calculated before the initiation of the experiment using the R package PowerUpR and the estimated effect size from previous studies. The minimum sample size was found to be n_min_ = 10. The calculated value for each CTI was an average of the 10 responses for that condition, except in the case of outliers (>2.5 standard deviations from the mean), which were excluded from the analysis (<0.01% of the data). Data from each CTI were compared to the average of 20 responses from the corresponding muscle when no movement was performed, where 10 responses were recorded prior to the start of the movement, and 10 responses were recorded after the completion of that movement. For each comparison, the normality of the difference values was tested with the Shapiro–Wilk test. If values were normally distributed (*p* > 0.05), a two-tailed paired sample Student’s t-test was used for analysis. If normality was violated, a Wilcoxon signed-rank test was performed. Significance set at *p* < 0.05 and a Bonferroni correction was applied to account for multiple comparisons. Second responses (R2) were removed if the first response (R1) produced a peak-to-peak value below 0.02 mV or where R2 > R1, which would suggest contamination from volitional muscle activation (<0.05% of the data). The R2/R1 ratio is given as the ratio between R2 and R1 for that pair of responses. Probability maps of the modulation of agonist, synergist, and antagonist motor pools during selected movements were constructed using the number of instances where a response for a given CTI and muscle was higher than any of the control responses and was divided by the total number of comparisons.

## 3. Results

### 3.1. Reaction Time

The average reaction time across movements during the auditory cue was determined to be 219.1 ± 41.8 ms, while the average reaction time across movements during the tactile cue was 200.9 ± 52.0 ms. The overall average reaction time for both cues across movements was 210.2 ± 47.7 ms. There were no significant differences (*p* < 0.05) in reaction time between movements or cue.

### 3.2. Auditory Condition

Figure 2 exemplifies evoked response changes for R1 and R2 for each muscle and movement task during the 200 ms CTI auditory condition for a representative participant. Changes in R1 can be seen in the agonist for each movement task when compared across tasks. Some movement tasks caused visible changes in contralateral responses, such as knee extension, which results in facilitation of the contralateral knee flexors (RMH), when compared to other movement tasks for that muscle.

Figure 3 presents the patterns of evoked response modulation during each movement preparation following auditory cues. Only muscle responses that were significantly different from control responses (*p* < 0.05) are shown (see Appendix A for all muscle responses). Muscles are listed along the bottom of each plot and the CTI tested are across the top. Data were normalized to the control condition (stimulation with no movement), and 100% of the average control response is equal to the distance between tick marks. The width of the connection represents the amount of facilitation/inhibition for a given muscle and CTI.

During knee extension movement using the auditory cue, we found R1 facilitation of all muscles for at least one CTI. The knee flexion movement produced R1 facilitation (top) of the agonist (LMH), the antagonist (LVL), LTA, RVL, RMH, RTA, and RSOL for at least one CTI. Dorsiflexion facilitated R1 responses for at least one CTI for all muscles apart from the antagonist (LSOL). The plantarflexion movement caused R1 facilitation of all muscles for at least one CTI. Facilitation of R2/R1 ratio (bottom) was found at the 100 ms CTI for most movements, apart from knee flexion, which first significance was found at the 50 ms CTI. Inhibition of the R1 response of the antagonist (LMH) was found during the knee extension movement at the 50 ms CTI. Inhibition of the R2/R1 ratio was found during knee flexion for the contralateral knee flexors (RMH) during the 100 ms and 150 ms CTIs. In general, the modulation of the evoked responses increased with increasing CTI duration up to the 250 ms CTI, for which double-pulse TSS was often delivered immediately after movement initiation (Figure 1C).

#### 3.2.1. First Responses (R1s)

Prior to knee extension, the largest facilitation of R1s occurred in the agonist (LVL). Facilitation of the response in LVL averaged 258% of control at 150 ms CTI. Facilitation increased during both 200 and 250 ms CTIs by 437% and 468%, respectively. R1s in contralateral knee flexors and extensors were also facilitated; responses in the RVL were 127% of control responses during the 100 ms CTI condition, with a maximum increase of 273% during the 250 ms CTI condition. In the RMH, a similar pattern of facilitation was observed, which first reached statistical significance at the150 ms CTI with 187%, compared to the control condition, and 246% during the 250 ms CTI. The antagonist (LMH) was the only muscle to show inhibition in the R1s, with an amplitude decrease of 85% at the 100 ms CTI.

Similarly, prior to knee flexion, the R1s in the agonist (LMH) increased in amplitude even after movement initiation at the 250 ms CTI (234%). In this case, facilitation of the antagonist (LVL) was 214% after movement initiation at the 250 ms CTI, and R1 in the contralateral knee extensor (RVL) was facilitated up to 192% at the 250 ms CTI.

Prior to dorsiflexion, facilitation of the R1s occurred in the agonist (LTA) at all CTIs with 121% facilitation at the 50 ms CTI, and it increased to 141%, 350%, 554%, and 556% during the 100, 150, 200, and 250 ms CTIs, respectively. No modulation of the R1 in the antagonist (LSOL) was observed. The first increase in LVL occurred at the 150 ms CTI with 137% and increased to 147% at the 250 ms CTI. The second largest evoked facilitation for the movement was measured from the RVL, which was 168% of the control response at the 250 ms CTI. Facilitation of the contralateral plantar flexor (RSOL) was similar, with 133% at the 250 ms CTI.

Prior to plantarflexion, R1s in the agonist (LSOL) were first significantly facilitated at the 100 ms CTI, with 110% of the control, and increased to 153%, 203%, and 215% for the 150, 200, and 250 ms CTIs, respectively. The largest facilitation was observed in the LVL for the condition, with significant facilitation of responses beginning at the 150 ms CTI (270%) up to the 250 ms CTI (437%).

#### 3.2.2. Second Responses (R2s)

The R2/R1 ratio increased prior to knee extension for the agonist (LVL) during the 100 ms CTI, with the R2/R1 ratio of 33.8% compared to the control value of 17.0%. The R2/R1 ratio in the contralateral knee flexor (RMH) increased to 24.2% for the same CTI. The R2/R1 ratio in the contralateral knee extensor (RVL) increased during the 200 ms CTI (31.1%) and 250 ms CTI (43.4%) when compared to the control ratio (21.3%). Prior to knee flexion, the R2/R1 ratio in the agonist (LMH) increased starting at the 100 ms CTI (43.2%) and was 30.5%, 22.2%, and 45.1% at the 150, 200, and 250 ms CTIs, respectively, while the control ratio was 2.6%. Prior to dorsiflexion, the R2/R1 ratio in the agonist (LTA) increased starting at the 100 ms CTI (15.2%) and remained significantly higher with a value of 55.2%, 57.0%, and 33.9% at the 150, 200, and 250 ms CTIs, while the control ratio was 8.9%. Prior to plantarflexion, the R2/R1 ratio in the agonist (LSOL) increased up to 20.6%, 22.2%, 11.5%, and 5.2% at the 100, 150, 200, and 250 ms CTIs, while the control ratio was 3.3%. The R2/R1 ratio in the antagonist (LTA) was also higher at the 100 and 200 ms CTIs (24.7% and 18.9%, respectively), while the control ratio was 11.6%.

### 3.3. Tactile Condition

Figure 4 exemplifies evoked response changes found in R1 and R2 for each muscle and movement task during the 200 ms CTI tactile condition for a representative participant. Facilitation of R1, and in some cases R2, can be seen for each movement when compared across movement tasks for individual muscles. For example, knee flexion caused facilitation of R2 for the agonist (LMH), while other movements had no visible R2 response.

Figure 5 shows the patterns of evoked response modulation during preparation for each movement following tactile cues. Only muscle responses that were significantly different from control (*p* < 0.05) are shown (see Appendix A for all muscle responses). During the tactile cue knee extension movement, we found R1 facilitation of the agonist (LVL), LTA, RVL, RMH, RTA, and RSOL for at least one CTI. The knee flexion movement produced R1 facilitation of the agonist (LMH), the antagonist (LVL), RVL, RMH, RTA, and RSOL for at least one CTI. Dorsiflexion during the tactile cue facilitated R1 responses for at least one CTI for all muscles. The plantarflexion movement also caused R1 facilitation of all muscles for at least one CTI. Facilitation of R2/R1 ratio was found at the 100 ms CTI for most movements, apart from knee flexion. No inhibition was found for the R1s or the R2/R1 ratio. In general, the modulation of the evoked responses increased with increasing CTI duration up to the 250 ms CTI, for which double-pulse TSS was often delivered immediately after movement initiation.

#### 3.3.1. First Responses (R1s)

Prior to the knee extension, R1s were facilitated for the agonist (LVL) as early as the 100 ms CTI, with a 111% increase when compared to control. The facilitation of the LVL response peaked at the 200 ms CTI, with a 335% increase, and after movement initiation was 265% for the 250 ms CTI. The maximum evoked facilitation for LTA was 313% at the 200 ms CTI and 257% for the 250 ms CTI. The earliest significant facilitation for contralateral knee flexor (RMH) was at the 100 ms CTI, which increased to 123% of control, then peaked at the 150 ms CTI with 190%, and remained consistent at 187% and 186% for both the 200 and 250 ms CTI. The largest facilitation for the contralateral side was for the RVL, which reached an evoked maximum of 206% increase at the 250 ms CTI.

The R1s prior to knee flexion followed a similar pattern. The agonist (LMH) response was facilitated at the 100 ms CTI and a 135% response increase. Maximum evoked facilitation for the antagonist (LVL) was at the 150 ms CTI (197%) and remained consistently high at the 200 and 250 ms CTIs, with 195% and 173% of control value, respectively. The similar facilitation was found for the LTA with an evoked maximum of 172% at the 200 ms CTI and, after movement initiation, was 141% at the 250 ms CTI. The greatest magnitude of facilitation for the contralateral side was the contralateral knee flexor (RMH) with an evoked maximum of 140% during the 150 ms CTI, while similar levels of facilitation were found for the contralateral knee extensor (RVL) with an evoked maximum of 137% at the 200 ms CTI.

Prior to dorsiflexion, the greatest magnitude of facilitation was evoked from the agonist (LTA) at the 200 ms CTI, which was 439% of control, and after movement initiation was 329% at the 250 ms CTI. On the contralateral side, the largest facilitation found was RMH at 133% during the 150 ms CTI, and 125% and 130% at the 200 and 250 ms CTIs, respectively. Similar levels of facilitation were measured in the RVL, with an evoked maximum of 131% at the 200 ms CTI, and after movement initiation was 127% at the 250 ms CTI.

Prior to plantarflexion, the agonist (LSOL) increased to 174% at the 200 ms CTI and, after movement initiation, was 163% at the 250 ms CTI. The largest ipsilateral activation occurred in the LVL, with an evoked maximum recorded facilitation at the 200 ms CTI, which was a 439% increase in amplitude, and after movement initiation was 313% for the 250 ms CTI. The largest contralateral response was observed in the RMH with an increase of 169% at the 200 ms CTI and 162% at the 250 ms CTI. The greatest magnitude of facilitation was measured in the RSOL (147%) and RTA (138%) at the 200 ms CTI.

#### 3.3.2. Second Responses (R2s)

The R2/R1 ratio in the LTA increased prior to knee extension at the 100 ms CTI (51.4%) and the 200 ms CTI (25.9%), whereas the control ratio was 7.6%. During preparation for knee flexion, the R2/R1 ratio increased in LTA only at the 150 ms CTI (30.1%) and the 200 ms CTI (37.4%), while the control ratio was 9.0%. Preparation for dorsiflexion caused changes in the R2/R1 ratio in the agonist (LTA) at the CTIs of 100, 150, 200, and after movement onset at 250 ms, with values of 31.8%, 46.0%, 51.4%, and 24.1%, while the control ratio was 9.8%. The R2/R1 ratio in the antagonist (LSOL) at the 200 ms CTI reached 10.3%, while the control ratio for the muscle was 3.2%. Prior to plantarflexion, the R2/R1 ratio in the LVL and RMH was increased at the 100 ms and 150 ms CTIs to 31.0% and 34.7%, while the control ratios were 19.4% and 4.6%, respectively.

### 3.4. Isometric Contraction Response

Figure 6 exemplifies evoked response changes found in R1 and R2 for each muscle and movement task during the isometric contraction condition for a representative participant. While significant R1 facilitation was observed, the magnitude was diminished when compared to the auditory and tactile conditions.

Figure 7 shows the patterns of evoked response modulation during isometric contraction tasks for each movement. Only muscle responses that were significantly different from control responses (*p* < 0.05) are presented (see Appendix A for all muscle responses). During isometric knee extension, we found R1 facilitation of the agonist (LVL) with inhibition of the LMH, LSOL, RVL, RMH, RTA, and RSOL. Isometric knee flexion produced facilitation of R1s in all muscles except for the RVL and caused inhibition in the LTA and LSOL. Isometric dorsiflexion facilitated R1 responses in all muscles, apart from the LMH, and caused inhibition of the LVL and LSOL. Isometric plantarflexion caused R1 facilitation in the agonist (LSOL) and antagonist (LTA) along with RVL, RTA, and RSOL. Plantarflexion did result in inhibition in any muscle tested. Facilitation of the R2/R1 ratio was found for most movements and at least one muscle apart from plantarflexion. Inhibition of the R2/R1 ratio was found during knee flexion for the RVL, dorsiflexion for the RVL and RSOL, and plantarflexion for RVL, RTA, and RSOL.

#### 3.4.1. First Responses (R1s)

Isometric knee extension caused the facilitation of R1s in only the agonist muscle (LVL) (169% of control). This was lower (*p* < 0.05) when compared to the 200 ms CTI in the auditory condition (437% of control). The movement caused the inhibition of the antagonist (LMH) (28% of control) and the LSOL (85% of control). All muscles on the contralateral side were inhibited. The contralateral knee flexor (RMH) was most inhibited, averaging 71% of the control. The contralateral knee extensor (RVL) had an R1 of 84% of control, while RTA and RSOL averaged 83% and 85%, respectively.

During isometric knee flexion, the agonist (LMH) was facilitated (132% of control), which was reduced (*p* < 0.05) when compared to the 200 ms CTI in the auditory condition, where facilitation reached 218% of control. Two muscles were inhibited: the LTA (78%) and the LSOL (74%).

Isometric dorsiflexion caused facilitation in the agonist (LTA) (221% of control) but was lower (*p* < 0.01) than during the 200 ms CTI in the auditory condition (554% of control). The movement also caused inhibition in the antagonist (LSOL) (49% of control).

Isometric plantarflexion caused facilitation in the agonist (LSOL) (117% of control), which was reduced (*p* < 0.01) compared to the 200 ms CTI auditory condition (203% of control) (an example of this can be seen in Figure 1C). The movement also caused facilitation in the agonist (LTA) (118% of control).

#### 3.4.2. Second Responses (R2s)

Isometric knee extension caused facilitation in the antagonist (LMH) with an average R2/R1 ratio of 27.8%, while the control ratio was 13.4%. During knee flexion, the agonist (LMH) facilitated with an average R2/R1 ratio of 21%, while the average control ratio was 10.3%. The contralateral antagonist (RVL) was inhibited, with a R2/R1 ratio decreasing to 32.1%, while the average control ratio was 59.5%. Isometric dorsiflexion caused facilitation in the agonist (LTA), which had an average R2/R1 ratio of 35.9%, while the control ratio was 28.6%. The R2/R1 ratio in the contralateral plantar flexor (RSOL) was decreased to 3.2% when compared to the control ratio which was 18.2%. During isometric plantarflexion the R2/R1 ratio in the contralateral plantar flexor (RSOL) decreased to 3.7%, while the control ratio was 27.4%. The contralateral dorsiflexor (RTA) was also inhibited with a R2/R1 ratio of 11.8%, while the ratio of control responses was 38.7%.

### 3.5. Probability Maps of Modulated Responses

To demonstrate temporal changes of the excitability of lumbosacral motor pools projecting to ipsilateral and contralateral knee and ankle flexors and extensors prior to and during the movements for the auditory, tactile, and isometric conditions, we created the probability maps of modulated R1s (Figure 8). Facilitation is shown using the muscle agonist color, and inhibition is shown using the muscle antagonist color.

#### 3.5.1. Knee Extension

During the auditory condition, the probability of R1 facilitation increased for the agonist (LVL) as CTI increased until it reached 100% facilitation at the 250 ms CTI. The antagonist (LMH) had a higher chance of inhibition up to the 150 ms CTI, but at the 250 ms CTI there was a higher (21%) probability of facilitation. The contralateral knee extensor (RVL) at the 250 ms CTI had 100% probability of facilitation. The contralateral knee flexor (RMH) had a higher probability of facilitation with a likelihood of 100% at the 250 ms CTI. Only at the 100 ms CTI was there a higher probability (13%) of inhibition.

During the tactile condition, the probability of facilitation peaked at the 150 ms CTI for the agonist (LVL) with a value of 98% and 87% at the 250 ms CTI. The antagonist (LMH) had a higher probability of facilitation, but it was less than 10% for all CTIs. The probability for facilitation of the contralateral knee extensor (RVL) also peaked at the 150 ms CTI with an 88% chance of facilitation. The maximum likelihood for facilitation of the contralateral knee flexor (RMH) was 96% occurring at the 150 ms CTI.

The isometric contraction condition had the largest differences between the agonist and antagonist muscles. The agonist had (LVL) a 100% chance of facilitation, while the antagonist (LMH) had a 100% chance of inhibition. The contralateral knee extensor (RVL) had a higher probability (27%) of inhibition, and there was a 100% chance of facilitation of the contralateral knee flexor (RMH).

#### 3.5.2. Knee Flexion

During the auditory condition, the probability of facilitation increased for the agonist (LMH) as the CTI increased with a maximum of 95% at the 200 ms CTI and was 88% at the 250 ms CTI. Facilitation increased for the antagonist (LVL) with a maximum of 97% at the 250 ms CTI. The contralateral knee flexor (RMH) had a higher probability of facilitation as CTI increased to a maximum at the 250 ms CTI with a 98% chance of facilitation. The contralateral knee extensor (RVL) had a maximum probability of facilitation of 65% at the 250 ms CTI.

During the tactile condition, the probability of facilitation of the agonist (LMH) reached a maximum of 88% at the 150 ms CTI. The antagonist (LVL) had a maximum of 57% probability for facilitation at the 150 ms CTI. The contralateral knee extensor (RVL) was found to have a higher probability (48%) of facilitation at the 150 ms CTI. The contralateral knee flexor (RMH) had a peak facilitation probability of 55% at the 150 ms CTI.

The isometric condition had a higher probability of facilitation in both the agonist (LMH) and the antagonist (LVL), with a likelihood of 48% and 65%, respectively. The contralateral knee flexor (RMH) and knee extensor (RVL) had a 53% and 32% probability for facilitation, respectively.

#### 3.5.3. Dorsiflexion

During the auditory condition, the agonist (LTA) had an increased probability of facilitation with a maximum of 96% at the 250 ms CTI. The antagonist (LSOL) had a more dynamic likelihood profile, with a slight probability of facilitation at the 50 and 100 ms CTIs (12% and 8%, respectively), but the 150 ms CTI had a 34% probability of inhibition, which dropped to 18% and 3% for the 200 and 250 ms CTIs, respectively. The contralateral dorsiflexor (RTA) reached a maximum of 76% probability of facilitation at the 250 ms CTI. The contralateral plantar flexor (RSOL) had an increased probability of facilitation as CTI increased, with a maximum of 60% chance of facilitation at the 250 ms CTI.

The tactile condition resulted in a higher probability of facilitation of the agonist (LTA), with a maximum of 73% at the 250 ms CTI. The antagonist (LSOL) probability was similar to what was found during the auditory condition, where there was a 31% chance of facilitation at the 50 ms CTI and 36% chance of inhibition at the 150 ms CTI. Both the contralateral dorsiflexor (RTA) and plantar flexor (RSOL) reached a maximum of 31% and 33% probability for facilitation at the 150 ms CTI, respectively.

During isometric contraction condition, the agonist (LTA) had a 99% chance of facilitation while the antagonist (LSOL) had a 100% chance of inhibition. The contralateral dorsiflexor (RTA) had a 99% chance of facilitation, and the contralateral plantar flexor (RSOL) had an 89% chance of facilitation.

#### 3.5.4. Plantarflexion

During the auditory condition, the probability of facilitation for the agonist (LSOL) increased as the CTI increased, with a maximum of 100% at the 200 and 250 ms CTIs. The antagonist (LTA) had a higher probability of facilitation as the CTI increased, with a maximum probability of 100% at the 250 ms CTI. The contralateral plantar flexor (RSOL) had a similar probability of response but had a maximum probability of facilitation at the 250 ms CTI of 86%, while the contralateral dorsiflexor (RTA) had a similar likelihood with a maximum of 87% probability of facilitation at the 250 ms CTI.

The tactile condition resulted in a maximum (74%) facilitation probability at the 150 ms CTI for the agonist (LSOL). The antagonist (LTA) had a higher probability of facilitation as the CTI increased to a maximum of 86% probability of facilitation at the 250 ms CTI. The contralateral plantar flexor (RSOL) had a higher probability of facilitation as the CTI increased, with a maximum of 78% at the 250 ms CTI. The contralateral dorsiflexor (RTA) also had a higher probability of facilitation as the CTI increased with a maximum of 77% probability for facilitation at the 250 ms CTI.

During isometric contraction, the probability of R1 facilitation for the agonist (LSOL) was 64%, while the antagonist (LTA) had a facilitation probability of 71%. The contralateral plantar flexor (RSOL) and dorsiflexor (RTA) were found to have a probability for facilitation of 98% and 100%, respectively.

## 4. Discussion

To better understand the descending modulation of spinal excitability prior to and during voluntary movements, we investigated the multi-segmental convergence of supraspinal inputs on the lumbosacral neural network. We demonstrated distinct changes in excitability of agonist and antagonist motor pools preceding voluntary movements, as well as during isometric contraction of the lower limb muscles, in neurologically intact individuals. Prior to the movements, the modulation of spinally evoked motor potentials was the most prominent in the primary agonists and antagonists, and this was also observed in other muscles. Across all muscles, the pattern of modulation of evoked responses was similar following either the auditory or tactile cues. During isometric contractions, the evoked responses were also modulated, but they were less pronounced and more agonist-antagonist specific. The second evoked responses (R2), which were delivered 50 ms following the first response (R1) and are inhibited at rest, were facilitated during the tested conditions and were the most prominent in the synergists. These results demonstrate generalized effects of the descending drive on lumbosacral motoneuronal and inter-neuronal circuitries.

### 4.1. Modulation of Spinally Evoked Motor Potentials Prior to the Movements

Our results demonstrate the net effects of the descending inputs on the excitability of ipsilateral and contralateral motor pools prior to movement, starting as early as 50 ms following an auditory or tactile cue. The corticospinal system is thought to be responsible for the regulation of excitability of agonist and antagonist muscles during initiation and maintenance of the voluntary motor tasks [42]. Prior studies have demonstrated the role of Ia inputs to motoneurons during movements, and they have shown that the presynaptic inhibition of Ia terminals can substantially reduce the size of monosynaptic Ia excitatory postsynaptic potentials in motoneurons [43]. These processes can also be modulated from motor centers in the brain [44]. In later research, Hultborn et al. (1987) confirmed the supraspinal origin of inhibition of the interneurons mediating presynaptic inhibition to Ia fibers terminating onto voluntarily discharging motoneurons. It has been also proposed that there is an inverse descending facilitation of the interneurons mediating presynaptic inhibition on Ia fibers to motoneurons innervating muscles that remain relaxed during a selective contraction [35]. The modulation of presynaptic inhibition seen in these experiments implies that there is a tonic presynaptic inhibition of Ia terminals at rest, controlled from supraspinal motor centers, and these descending influences are removed or affected following spinal lesion [39,45]. Thus, it seems likely that the facilitation we observed prior to the movements was the result of the descending downregulation of inhibitory interneurons (disinhibition), rather than the direct excitation of specific motor pools. This arrangement allows corticospinal projections to the lumbosacral spinal cord to function as a substrate for the coordinated and automatic control of the lower limbs with incorporation of ongoing afferent input, rather than directly relaying the cortical commands on specific motor pools during each individual movement (e.g., steps). The distinct physiological and functional role of the corticospinal projections in the modulation of the lumbar sensorimotor network was demonstrated in rodents [46]. Specifically, it has been proposed that, at lumbar level, the main role of the corticospinal tract is the modulation of sensory inputs, which is a critical component of the selective regulation of sensory feedback used to ensure well-coordinated movements of the hindlimbs. In contrast, in the forelimbs, the corticospinal tract can directly induce motor contractions, independently of its supraspinal collaterals, through spinal targets located at the cervical level. Similar distinction in the function of the corticospinal system projecting to the cervical and lumbar spinal cord has also been proposed in primates [47]. Although the organization of the corticospinal tract undoubtedly differs between species, our findings support the previous works and likely provide the basis to characterize the function and role of descending regulation of spinal sensorimotor networks.

### 4.2. Modulation of Spinally Evoked Motor Potentials during Voluntary Contractions

Previous studies have reported the inhibition of spinally evoked motor potentials during passive muscle stretching and the facilitation of the responses during voluntary upper limb muscle contractions [48]. Modulation of evoked potentials in the lower limb muscles during voluntary contractions has also been studied [37,49,50], and, overall, it demonstrates the facilitation of agonists and the inhibition of antagonists. At the same time, the interplay between the ipsilateral and contralateral agonists and antagonists remains unclear. For instance, Minassian et al. (2007a) demonstrated the inhibition of evoked potentials in TA, SOL, and MH during the dorsiflexion and facilitation of ipsilateral SOL and VL during plantarflexion; on the contralateral side, only VL was facilitated during both plantarflexion and dorsiflexion, and the responses from the rest of the muscles were unchanged compared to control. Hofstoetter et al. (2008) demonstrated the facilitation of evoked response responses in ipsilateral TA and VL and inhibition in SOL and MH during dorsiflexion; whereas changes during plantarflexion were not significant in any muscle (note, however, that the sample size was small in this study with *n* = 3). Saito et al. (2021) found inhibition in MH during dorsiflexion and facilitation in SOL during knee extension. In our experiments, during isometric knee extension and dorsiflexion, we observed facilitation in the agonists (i.e., in VL and TA, respectively) and inhibition of the antagonists (i.e., in MH and SOL, respectively); whereas, during isometric knee flexion and plantarflexion, both the agonist and antagonist were facilitated. On the contralateral side, isometric knee flexion, plantarflexion, and dorsiflexion resulted in the facilitation of the evoked responses in the majority of muscles, whereas isometric knee extension caused inhibition of the responses in all recorded contralateral muscles (Figure 7). The inconsistency between the previous and present studies can be attributed to the functional flexibility of the spinal circuitry, required to produce movements involving many different potential combinations of muscle contractions, as well as spinal mechanisms triggered by specific contracting muscles [50], including heteronymous recurrent inhibition originated from the agonists and affecting the circuitry of synergists [51], inter-segmental inhibitory force feedback [52,53], and presynaptic mechanisms [35]. Differences in the motor tasks, fatigue, lower limb joints’ angle position, number of joints involved in the movement, and posture of the study participants in the above studies could contribute to the inconsistent findings. These considerations diminish the value of using maintained muscle contractions during spinal stimulation as the experimental paradigm to quantify the function and activity of supraspinal connections and spinal inter-neuronal network. In addition, motor tasks requiring the research subject to maintain individual muscle contractions may not be feasible for clinical populations with severe neurological impairments, such as SCI.

### 4.3. Probability of Modulated Responses in Agonists and Antagonists

Our results demonstrate temporal changes in the excitability of motoneurons prior to movement onset, where the probability of facilitation of the muscle agonist peaks at 150 to 200 ms following the auditory or tactile cue, which is about 100 to 50 ms prior to the onset of contraction. We found that a high probability of inhibition in antagonists occurred prior to and during dorsiflexion and knee extension, while a high probability of co-facilitation in antagonists occurred prior to and during plantarflexion and knee flexion (Figure 8). The difference in modulation of excitability in antagonists during different movements can be explained by the functional role of the muscles executing the movement. For instance, during standing, the anteriorly located dorsiflexors and knee extensors counteract posterior shifts of the body center of mass and provide fast and accurate postural adjustments, thus requiring more dynamic and “fine-tuned” coordination between the agonists and antagonists. Conversely, the posteriorly located plantar flexors and knee flexors continuously resist the gravity-induced torque pulling the body forward, and lower leg joint stability can “gain” from co-contracted antagonists [54,55,56]. Similarly, during stepping, co-facilitation of the ipsilateral agonist-antagonist couple during plantarflexion, occurring in the stance phase of the gait cycle, would increase ankle stiffness thus improving the joint stability. It is worth noting that the manifest difference in the recruitment rate of evoked responses of the anteriorly and posteriorly located leg muscles was also observed previously when examining the maximal slope of their recruitment curves [22], indicating different tonic level of presynaptic inhibition of Ia terminals and distinct neurophysiological properties of the circuitry in the leg muscles.

### 4.4. Modulation of Second Response Prior to and during the Movements

We sought to investigate post-activation depression in lumbosacral motor pools prior to and during movement initiation as another measurement of the functional state of corticospinal and spinal inter-neuronal networks in an intact spinal cord. We suggested that, in addition to being modulated as a function of the motor state, the dynamic of the R2 recovery in different muscles would differ between conditions and movements, given that a spinal stimulus results in bilateral, multi-segmental activation of lumbosacral roots. A reduction of the evoked potential amplitude, in a previously activated sensory-motor synapse attributed to post-activation- or homosynaptic-depression, is caused by a combination of presynaptic [57] and postsynaptic [58] inhibition mechanisms. Studies using the H-reflex demonstrated that repeated stimulation inhibits spinal motoneurons, likely because of their afterhyperpolarization [59], recurrent inhibition [60], and changes in the Ia afferent terminal leading to a transient reduction in neurotransmitter release [61,62,63]. Similar to the H-reflex, spinally evoked motor potentials exhibit post-activation depression in that the second pulse delivered within the next 20–2000 ms elicits a smaller response [39,40,41]. Post-activation depression can be reduced when the agonist muscle is voluntarily activated [40,64,65]. The mechanisms of this reduction are not well understood and are attributed to an increase in the neurotransmitter release due to increased frequency of impulses fired by muscle spindles in the contracting muscle [61].

We demonstrated, for the first time, the dynamics of the second response (R2) modulation in different leg muscles prior to and during voluntary movements. The most prominent changes in reduction of post-activation depression in our experiments were observed prior to the movement, when the R2/R1 ratio increased up to 51.4% at the CTIs between 100 and 200 ms in quiescent agonist muscles. These findings are coherent with previous observations that the ongoing descending drive decreases the amount of presynaptic inhibition of Ia fibers acting on spinal motoneurons [66,67].

### 4.5. Clinical Implications

Traditionally, rehabilitation clinicians have utilized subjective neuromotor assessments in a limited set of muscles for the determination of spared motor function, which is suggested to reflect the function of descending tracts and confer rehabilitative prognosis after SCI [68,69,70,71]. Among the descending tracts, the corticospinal system is a compelling target that is not only “neurophysiologically testable” [7,27,28,42] but is also one of the major biomarkers of neuromuscular recovery after SCI [72,73]. A key limitation of the existing clinical assessment tools is the inability to quantify residual corticospinal connectivity, whose influence on spinal networks is insufficient to produce muscle activation within the constraints of the examination [4,5,6,31,74,75]. It is remarkable that in our study, the observed multiplex changes of spinal motoneuronal excitability were occurring prior to the actual movement. This not only reflects the level of function of corticospinal and spinal inter-neuronal networks, but it also allows clinicians and researchers to perform objective neuromotor assessment even in the absence of visible voluntary contractions.

Furthermore, our findings on the interaction of the descending drive to the lumbosacral sensorimotor network and spinal stimulation suggest that disinhibition within spinal networks, occurring at the onset of voluntary efforts, may be critical in the context of the administration of neuromodulatory approaches using repetitive spinal stimulation administered to regain motor function in clinical populations. Currently, the dominating hypothesis in the field is that “artificial” afferent inputs, induced by epidural or non-invasive electrical stimulation of the spinal cord, can elevate the functional state of excitability of spinal networks, re-enabling or augmenting voluntary movement functions [8,14,16,23,27]. Conversely, it has been demonstrated that the repetitive electrical stimuli applied to the spinal cord at rest enhances intrinsic inhibitory mechanisms within the sensorimotor networks, as revealed by reduced spasticity following ESS [76,77] or TSS [78,79]. Based on these observations, one can propose that the excitatory or inhibitory effects of spinal stimulation on neuromotor outcomes depend on the presence or absence of descending drive to the spinal circuitry. Our data demonstrate that, prior to and during voluntary isometric contractions, the excitability of motor pools projecting to agonists and antagonists, as well as to the contralateral muscles, is, in general, elevated. Thus, it can be proposed it is the descending drive (residual in case of incomplete SCI) that in fact elevates the state of spinal inter-neuronal network excitability, while electrical spinal stimuli target specific populations of inter- and/or motoneurons and serve to regain given functions. Indeed, it is known that ESS and TSS demonstrate different patterns of specificity in activating selected motor pools [15,22,80,81,82], likely due to the different local and wide field stimulation capacity. To the best of our knowledge, every study utilizing ESS during voluntary effort after motor complete SCI has been successful so far in regaining muscle-specific control below the lesion in the presence of stimulation [14,15,82,83,84], while the fine and selective voluntary activation of specific agonists (with minimum co-contraction of antagonists) below the spinal lesion has yet to be shown with TSS. These observations may challenge or at least refine the current view on the enhancing vs. inhibitory effects of spinal stimulation on spinal excitability, if applied alone and without voluntary effort or activity. Whether our current findings change the existing activity-based neuromodulation approaches in neuro-restoration is a question for further research in clinical populations; however, this underlines the significance of proper and likely individual selection of the stimulation location, intensity, frequency, and, most importantly, timing between voluntary effort and spinal stimuli, to target specific movements or functions.

Changes in the functional state of excitability of spinal neuronal networks due to neuromodulation have been proposed as the primary mechanism of motor recovery following SCI [8,14,23,27,28]. The term “functional state” of spinal neuronal excitability can be reflective of the resting membrane potential of motoneurons, interneurons, as well as their responsiveness to afferent inputs and descending drive. Although all the above factors most likely play a role in neurophysiological and functional effects during neuromodulation, their quantitative characterization is required. Without that, attributing the effects to “changes in functional state” remains vague and difficult to reproduce in large sample size trials. We propose that the spatiotemporal characterization of excitability in response to descending commands provides the way to characterize the functional state of spinal sensorimotor networks. Investigation of the relationship between the degree of remaining supraspinal connections with spinal networks and the ability to voluntarily control muscles of the lower limbs in individuals with SCI will be the next step to quantify the residual and regained function throughout the course of neurorehabilitation. Finally, the demonstrated feasibility of evaluating spinally evoked motor potentials’ modulation during voluntary movement attempts following the auditory and tactile cues offers opportunities to probe specific sensorimotor pathways in individuals with various neurological disorders and injuries, including SCI, stroke, multiple sclerosis, hereditary spastic paraplegia, and various forms of peripheral neuropathy.

## 5. Conclusions

We demonstrated the modulation of the excitatory state of spinal sensorimotor networks in response to descending commands during both the preparatory stage and execution of voluntary movements. Observed changes in spinal motoneuronal excitability during our experiments were likely triggered by presynaptic and postsynaptic mechanisms. Movement-specific modulation of motoneuronal disinhibition within spinal networks was prominent and consistent prior to the actual movements. Disinhibition of ipsilateral and contralateral agonists and antagonists indicate that the corticospinal projections to the lumbosacral spinal cord serve an essential, yet indirect, component of the selective tuning of motor output based on sensory feedback to ensure coordinated and patterned movement. During isometric muscle contractions, the afferent discharge occurring from the contracting muscles influences the regulation of inter-neuronal networks, and further modulates the agonist-antagonist activation. The spatiotemporal characterization of multi-segmental motoneuronal excitability in response to descending commands provides the way to characterize the functional state of spinal sensorimotor networks. The excitation of spinal sensorimotor networks caused by the descending drive can be further manipulated by spinal stimulation targeted to specific motor pools to enable a desired motor output.

## Figures and Tables

**Figure 1 jcm-10-05958-f001:**
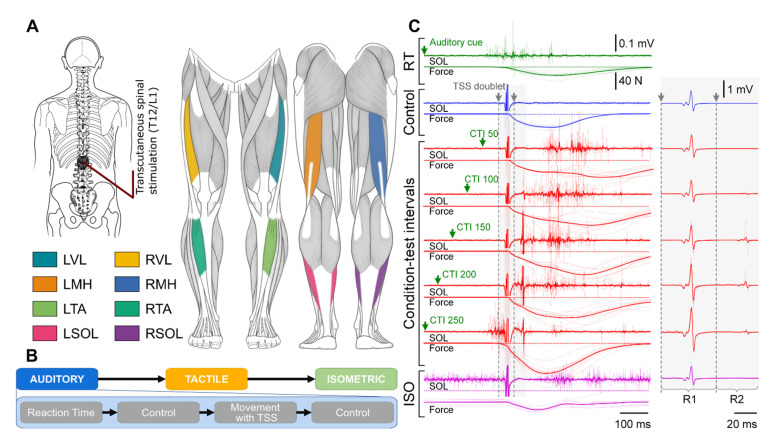
Experimental setup: transcutaneous spinal stimulation (**A**) was applied midline between the T12 and L1 spinous processes. EMG responses were recorded bilaterally from the vastus lateralis (VL), medial hamstring (MH), tibialis anterior (TA), and soleus (SOL) muscles. (**B**) Three experimental conditions were used during each movement task: an auditory cue, a tactile cue, and isometric contraction while TSS doublets were applied. (**C**) All CTIs are shown in red for the left SOL during the plantarflexion task for the auditory condition. Green designates reaction time (RT), blue designates control, and purple designates isometric contraction (ISO). The left and right panels use different time and amplitude scales to better visualize the waveform for voluntary SOL activation during plantarflexion (left panel) and R1 and R2 responses (right panel). Arrows indicate each pulse during the TSS doublets. The dotted horizontal line marks the initial force level prior to movement (left panel). Individual traces are shown with thin lines, and the average response is designated with a bold line.

**Figure 2 jcm-10-05958-f002:**
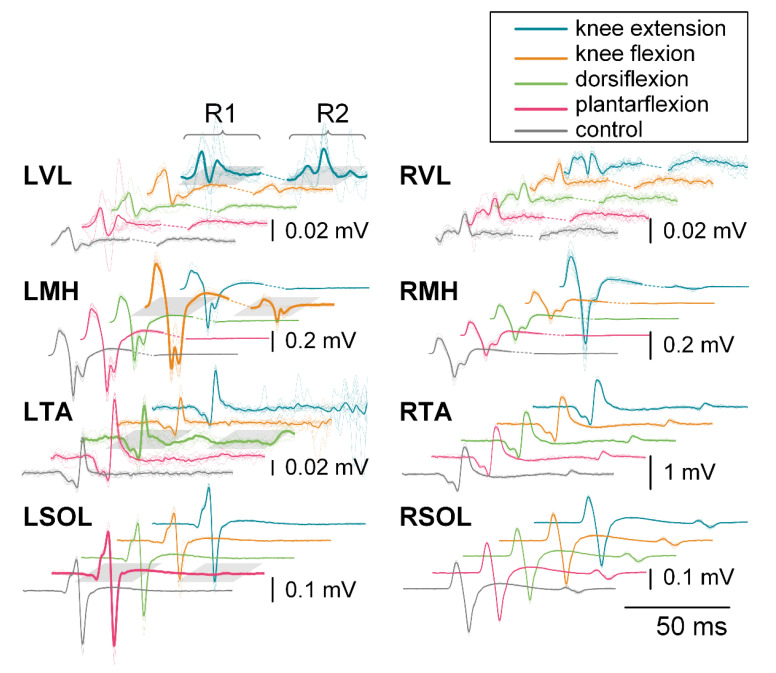
R1 and R2 responses for each muscle during knee extension (blue), knee flexion (orange), dorsiflexion (green), plantarflexion (pink), and control (gray) during the 200 ms CTI auditory condition for a representative participant. Individual responses are traces represented with thin dotted lines, while the average is shown with a thin solid line. The bold line signifies the agonist for that movement. Different amplitude scales are used for each muscle to better visualize the waveforms.

**Figure 3 jcm-10-05958-f003:**
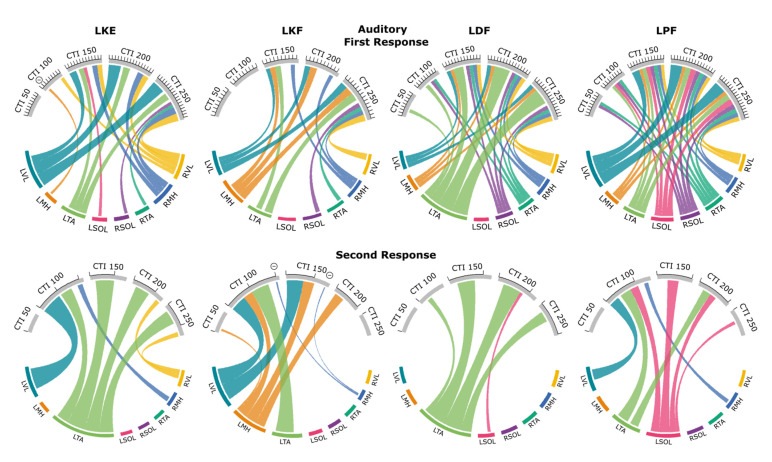
Excitability and inhibitory interactions during left knee extension (LKE), left knee flexion (LKF), left dorsiflexion (LDF) and left plantarflexion (LPF) movements using auditory cues: averaged significant (*p* < 0.05) facilitations or inhibitions during first (**top**) or second (**bottom**) responses during auditory cues. Each first response is scaled to the control for that movement condition, where the average control response is 100% and corresponds to the distance between tick marks. Second responses are given as the ratio between the second and first response for that pair of responses. Individual muscles are grouped along the bottom of each plot. Thickness of connections denotes the amount of inhibition or facilitation found where significant inhibition is designated with a circled minus sign.

**Figure 4 jcm-10-05958-f004:**
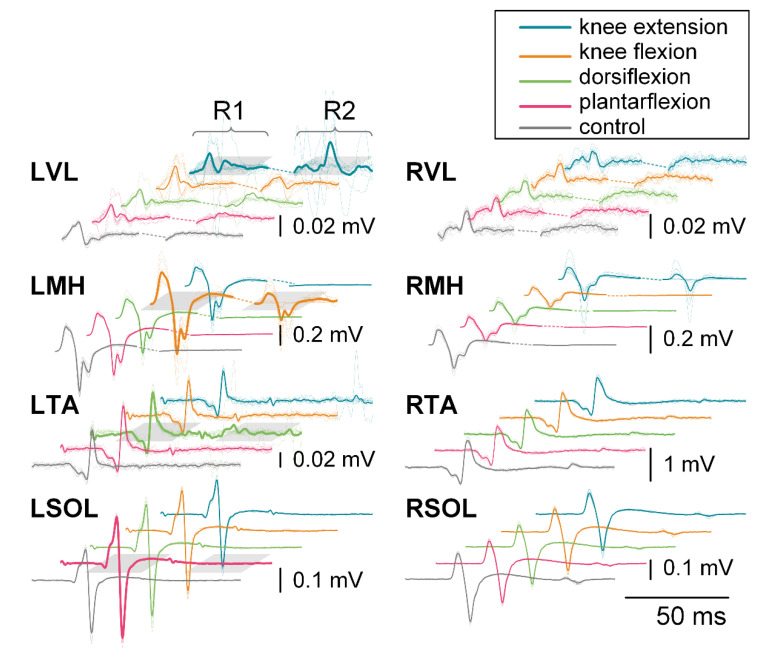
R1 and R2 responses for each muscle during knee extension (blue), knee flexion (orange), dorsiflexion (green), plantarflexion (pink), and control (gray) during the 200 ms CTI tactile condition for a representative participant. Individual responses are traces that are represented with a thin dotted line, while the average is shown with a thin solid line. The bold line signifies the agonist for that movement. Different scales are used for each muscle to better visualize the waveforms.

**Figure 5 jcm-10-05958-f005:**
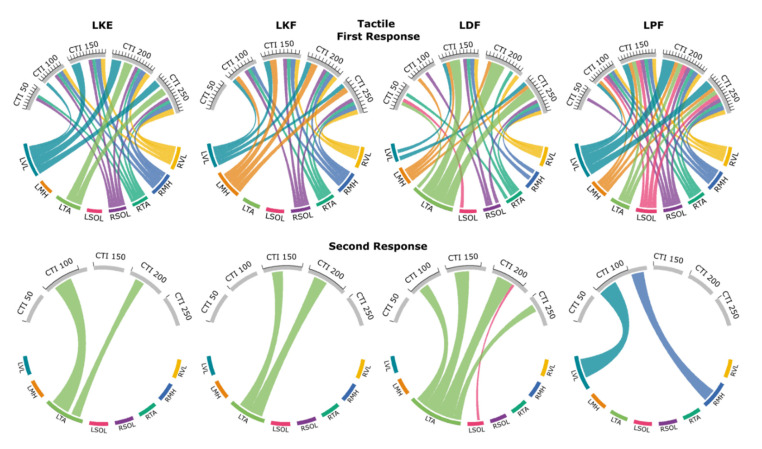
Excitability and inhibitory interactions during left knee extension (LKE), left knee flexion (LKF), left dorsiflexion (LDF) and left plantarflexion (LPF) movements using tactile cues: averaged significant (*p* < 0.05) facilitation or inhibition during first (**top**) or second (**bottom**) responses during tactile cues. Each first response is scaled to the control condition where control is 100% and corresponds to the distance between tick marks. Second responses are given as the ratio between second and first response for that pair of responses. Individual muscles are grouped along the bottom of each plot. Thickness of connections denotes the amount of inhibition or facilitation found where significant inhibition is designated with a circled minus sign.

**Figure 6 jcm-10-05958-f006:**
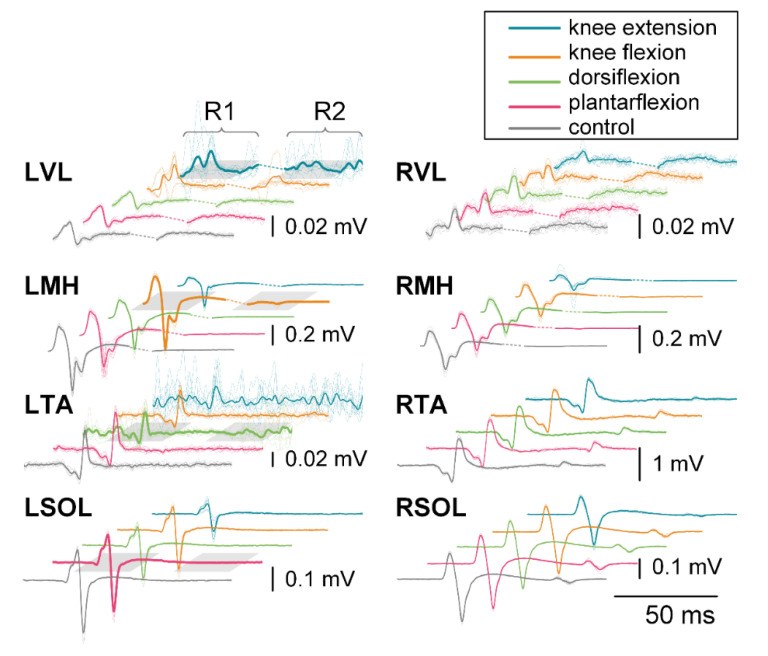
R1 and R2 responses for each muscle during knee extension (blue), knee flexion (orange), dorsiflexion (green), plantarflexion (pink), and control (gray) during the isometric contraction condition for a representative participant. Individual responses are traces that are represented with a thin dotted line, while the average is shown with a thin solid line. The bold line signifies the agonist for that movement. Different scales are used for each muscle to better visualize the waveforms.

**Figure 7 jcm-10-05958-f007:**
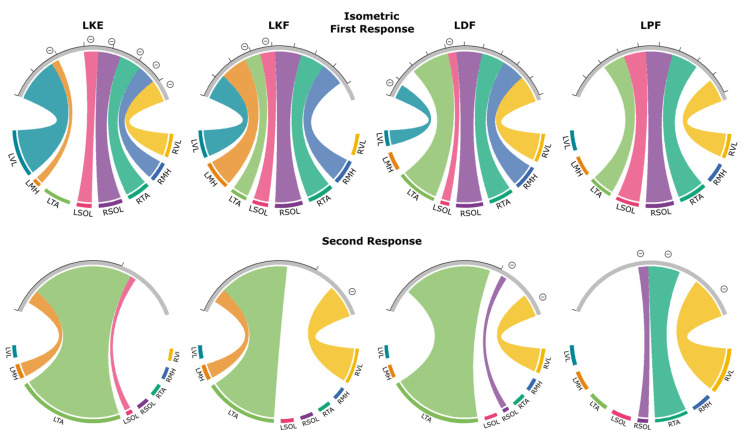
Excitability and inhibitory interactions during left knee extension (LKE), left knee flexion (LKF), left dorsiflexion (LDF) and left plantarflexion (LPF) isometric contraction: averaged significant (*p* < 0.05) facilitations or inhibitions during first (**top**) or second (**bottom**) responses during tactile cues. Each first response is scaled to the control condition where control is 100% and corresponds to the distance between tick marks. Second responses are given as the ratio between second and first response for that pair of responses. Individual muscles are grouped along the bottom of each plot. Thickness of connections denotes the amount of inhibition or facilitation found where significant inhibition is designated with a circled minus sign.

**Figure 8 jcm-10-05958-f008:**
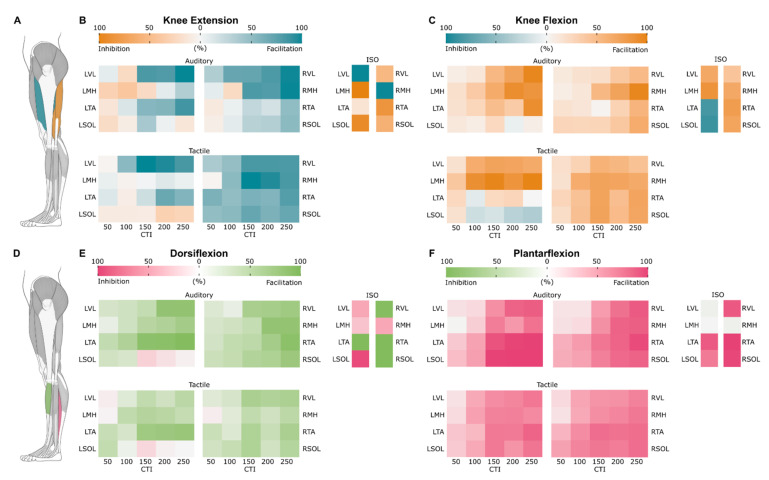
Motoneuronal excitability map: temporal changes in excitability during (**B**) left knee extension (LKE), (**C**) left knee flexion (LKF), (**E**) left dorsiflexion (LDF), and (**F**) left plantarflexion during auditory and tactile conditions, as well as isometric contraction. Each row represents one muscle, and each column represents one CTI. Facilitation and inhibition are denoted using the muscle agonist and antagonist colors, respectively (**A**,**D**). Note that a high probability of inhibition in antagonists during knee extension and dorsiflexion, as well as a high probability of co-facilitation in antagonists during plantarflexion and knee flexion, occurred across conditions.

## Data Availability

The data presented in this study are available on request from the corresponding author.

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
