# Peer review of "Characterization of Spinal Sensorimotor Network Using Transcutaneous Spinal Stimulation during Voluntary Movement Preparation and Performance"

_jcm, 2021, doi:10.3390/jcm10245958_

Round 1

Reviewer 1 Report

The mail goal of the reviewed manuscript was to investigate the multi-segmental convergence of the descending drive on the lumbosacral neural networks in healthy subjects using double pulse of transcutaneous electrical stimulation. The field of the study is interesting and original. The manuscript seems to be well written, and the methodology is well conducted. The hypothesis is well explaining in the introduction. My comments are more meant to consider enhancing the topic for the readership rather than specifically review the methodology of the study:

I recommend using the concept “transcutaneous spinal cord stimulation (tSCS)” rather than Trasncutaneous electrical spinal stimulation (TSS). The first one is most used in the scientific literature.

I suggest including “transcutaneous spinal cord stimulation” or “spinal stimulation” in the key words of the manuscript

Last paragraph of the introduction: “…stimulation delivered over the posterior roots entering the spinal cord can generate evoked potentials in multiple proximal and distal leg muscles bilaterally and simultaneously.” This is the called the posterior root-muscle (PRM) reflex, isn`t it? If no, please explain de difference

Data about the included studies (sex, age, height and weight) is recommended in the results section.

Why do you use a double pulse and not single pulses? Please explain it to the readers

Statistical Analysis: “…average of the 10 responses for that condition except in the case of outliers (> 2.5 standard deviations from the mean)”. ¿Could you include in the results section how many outliers (or an idea of the proportion) were found?

I don`t understand the figures 3, 5 and 7. I think that this format is very complex for the readers. Please consider other formats or use a table.

Reviewer 2 Report

These authors have submitted the manuscript entitled ‘Characterization of spinal sensorimotor network using transcutaneous spinal stimulation during voluntary movement preparation and performance’ as a research article, and they performed transcutaneous spinal stimulation (TSS) to T12-L1 vertebrae, and found the inter-neuronal circuitry was modulated in normal subjects. This manuscript is interesting, but there are some major points need to be answered for the publication to Journal of Clinical Medicine.

  1. These authors already wrote an article about the effects of unilateral TSS to intact controls (doi:10.1152/jn.00454.2019), and following article about the effects of TSS and epidural spinal stimulation (ESS) to spinal cord injured subjects, which was published to JCM (doi:10.3390/jcm10214898) recently. Basically, electrical currents spread throughout back area of the body (volume conduction) when using transcutaneous electrodes, therefore there might be no significant difference in the amount of electricity transmitted on the midline or lateral side of the back although back muscles close to the electrode may be stimulated differently as shown in a review article (doi:10.1371/journal.pone.0260166). In addition, the mechanism of the activation of spinal sensorimotor networks would not be different between midline and lateral stimulation of transcutaneous electrode on the back.
  2. These authors placed wireless surface electrodes on 4 selected muscles at each side. As shown in Fig. 1A, soleus muscle is placed under gastrocnemius muscle and tendon, and muscle area closest to the surface is separated to medial and lateral sides from Achilles tendon, therefore soleus muscle actually hard to be recorded using one surface electrode, and recorded signals when a subject perform ankle plantar flexion might be from gastrocnemius muscle as well as soleus muscle. Therefore, recording medial or lateral gastrocnemius is more appropriate than recording soleus, and many of previous studies had used gastrocnemius muscles (doi:10.1371/journal.pone.0260166).
  3. In the discussion, they wrote “Across all muscles, the pattern of modulation of evoked responses was similar following either the auditory or tactile cues; however, the facilitation of ipsilateral VL and MH seemed to occur earlier following the tactile cues.”, and they did not explain why auditory and tactile cues showed different results. All subjects in this study received auditory cue prior to tactile cue as shown in figure 1B. Since the experiment was conducted in this order, they should check whether there is a possibility that some kind of learning effect has appeared.
  4. I wonder whether results would be different when subjects moved their right knee and ankle, and why these authors performed left leg only in this experiment.
  5. They used various condition-test intervals (50, 100, 150, 200, and 250 ms) after auditory or tactile cue, but R1 and R2 responses were displayed only during 200 ms CTI in figure 2 and 4. They might explain why they did not show all results according to different CTI.

Reviewer 3 Report

This is a neurophysiological study investigating supraspinal modulation of lumbosacral neural networks before and during voluntary leg movements. Thirteen healthy subjects (HS) underwent an experimental protocol consisting of transcutaneous spinal stimulation sessions over the T12-L1 vertebrae while performing isometric leg movements (i.e., knee flexion and extension, plantar flexion and dorsiflexion). Also, transcutaneous spinal stimulation was delivered within specific time intervals (i.e., 50 to 250 ms) after an auditory or tactile stimulus aimed to trigger movement initiation in HS (accordingly spinal stimulation was delivered during movement preparation). The authors demonstrated changes in the spinal motoneuron excitability by showing facilitation of evoked motor potentials in specific muscle patterns following experimental procedures. Overall, the findings of the study help clarify the effects of supraspinal modulation on inter-neuronal spinal circuitry responsible for the coordinated and patterned movement of the lower limbs. Moreover, the authors also examine possible clinical perspectives.

The topic of the study is of interest. Also, methods and experimental procedures are well described.  The manuscript is clear and well written. Accordingly, I have only a few minor comments for the authors before publication of the paper:

- Taking into account specific pathological conditions associated with isolated corticospinal tract degeneration of the lower limbs, such as pure forms of hereditary spastic paraparesis (e.g., SPG4), would add value to the paper. Please, consider discussing this perspective.

- The methods should better clarify the inclusion criteria of the study and enrolling procedures.

- The manuscript is rather verbose. Accordingly, the authors should make an effort to reduce the total word count to improve the readability of the text.

- The low sample size of enrolled subjects is a possible limitation to be discussed.

Round 2

Reviewer 2 Report

These authors have responded well to a reviewer's comments, and I think this manuscript is ready to be published to JCM.